# Incidence of sinus thrombosis with thrombocytopenia—A nation-wide register study

Petteri Hovi[1,2]*, Arto A. Palmu[1], Tuomo A. Nieminen[3], Miia Artama[4,5], Jukka Jokinen[3], Esa Ruokokoski[3], Riitta Lassila[5,6,7], Hanna Nohynek[5], Terhi Kilpi[8]

1 Public Health and Welfare, Finnish Institute for Health and Welfare, Helsinki, Finland, 2 Pediatrics, Children's Hospital, Pediatric Research Center, University of Helsinki and Helsinki University Hospital, Helsinki, Finland, 3 Knowledge brokers, Finnish Institute for Health and Welfare, Helsinki, Finland, 4 Faculty of Social Sciences, Tampere University, Tampere, Finland, 5 Health security, Finnish Institute for Health and Welfare, Helsinki, Finland, 6 Coagulation Disorders Unit, Department of Hematology, Comprehensive Cancer Center, Helsinki University Hospital, Helsinki, Finland, 7 Research Program Unit in Systems Oncology, Faculty of Medicine, University of Helsinki, Helsinki, Finland, 8 Management, Finnish Institute for Health and Welfare, Helsinki, Finland

* petteri.hovi@thl.fi

**Data Availability Statement:** A minimal dataset is available in Table 1 in the manuscript. It displays the event counts and incidences for each

## Abstract

Safe vaccination is essential for mitigation of the COVID-19 pandemic. Two adenoviral vector vaccines, ChAdOx1 nCov-19 (AstraZeneca) and Ad26.COV2.S (Johnson&Johnson/Janssen) have shown to be effective and they are distributed globally, but reports on serious cerebral venous sinus thrombosis (CVST) associated with thrombocytopenia, have emerged. Our objective was to evaluate the background incidence of CVST with thrombocytopenia and to compare it to incidences following COVID-19 vaccines. We conducted a register-based nation-wide cohort study in Finland, including all 5.5 million individuals alive in Finland, 1 Jan 2020. COVID-19 vaccinations registered in the National Vaccination Register served as the exposure. We detected CVST admissions or hospital visits recorded in the hospital discharge register from Jan 1, 2020 through April 2, 2021. We confirmed the diagnosis of CVST and thrombocytopenia (platelet count <150,000 per cubic millimeter) using radiology reports and laboratory data. By Poisson regression, we compared the baseline incidences to the risks within four weeks after COVID-19 vaccinations. Out of the 167 CVST episodes identified in the registers, 117 were confirmed as CVST, 18 of which coincided with thrombocytopenia (baseline incidence 0.18 per 28 days per million persons). We found 2 episodes of CVST with thrombocytopenia within 28 days of the first ChAdOx1 nCov-19 vaccination (among 200,397 vaccinated, aged 16 or above). No cases were found following the first mRNA vaccine dose among 782,604 vaccinated. The background incidence of CVST combined with thrombocytopenia was minuscule compared to the incidence during the weeks following the ChAdOx1 nCov-19 vaccination. Accurate estimation of the baseline incidence is essential in the critical appraisal of the benefit-risk of any vaccination program.

combination of age-group, sex, and exposure status.

**Funding:** The current study received no external funding outside THL and there are no conflicting interests. Finnish Institute for Health and Welfare (THL) conducts Public-Private Partnership with vaccine manufacturers and has received research funding from Sanofi Inc., Pfizer Inc., and GlaxoSmithKline Biologicals SA. A.A.P., T.N., J.J., E.R., and T.K. have been investigators in these studies, but they have received no personal remuneration. P.H. reports being, in 2016 and 2017, a member in an expert group "Future Rheumatology Advisory Board", funded by Pfizer Inc.

**Competing interests:** The current study received no external funding outside THL and there are no conflicting interests. Finnish Institute for Health and Welfare (THL) conducts Public-Private Partnership with vaccine manufacturers and has received research funding from Sanofi Inc., Pfizer Inc., and GlaxoSmithKline Biologicals SA. A.A.P., T.N., J.J., E.R., and T.K. have been investigators in these studies, but they have received no personal remuneration. P.H. reports being, in 2016 and 2017, a member in an expert group "Future Rheumatology Advisory Board", funded by Pfizer Inc. This does not alter our adherence to PLOS ONE policies on sharing data and materials. However, by Finnish law, the authors are not permitted to share individual-level register data.

## Introduction

The COVID-19 pandemic caused by the coronavirus SARS-CoV-2 has by December, 2021, contributed to at least 271 million cases and 5.3 million deaths [1]. Vaccines are crucial in the mitigation of the pandemic. COVID-19 vaccine ChAdOx1 nCov-19 (AstraZeneca), is an adenoviral vector coronavirus vaccine conditionally approved by medical authorities throughout the world, introduced in the UK and other European countries and internationally via the COVAX facility in early 2021. Spontaneous reports to the marketing authorisation holders or the national regulatory authorities of several countries, including Denmark, Norway [2], Germany and Austria [3], UK [4, 5], and Australia [6], revealed several cases of thrombosis with thrombocytopenia syndrome [7] (TTS) following vaccination with the ChAdOx1 nCov-19. A great number of studies followed [8] (S1 Table).

On March 18, 2021, the European Medicine Agency (EMA) announced that although the benefits of the ChAdOx1 nCov-19 vaccine outweigh the risk of adverse effects, both vaccinees and health care professionals should be alerted about the very rare, unusual blood clotting accompanied by thrombocytopenia [9].

The affected patients suffered from a new syndrome that resembled the heparin-induced thrombocytopenia [10]. The most common reported site of thrombosis was the cerebral venous sinus, with some cases occurring also in other rare locations, including the splanchnic and portal veins [2–7, 11]. Thrombocytopenia has been the key observation in the TTS cases observed after covid-19 adenovector vaccination. However, the current knowledge of the background incidence of the combined thrombocytopenia and CVST is based on hospital data [12]. We performed a nation-wide register-based cohort study in Finland to estimate the coincidence of CVST and thrombocytopenia and the risks after COVID-19 vaccinations.

## Methods

We aimed at identifying all definitive cases of thrombotic thrombocytopenic syndrome (Level 1, Interim Brighton Criteria) [7] with CVST. This means new-onset thrombocytopenia with platelet count less than 150,000 per cubic millimeter and CVST confirmed by appropriate imaging techniques [7].

### Study population

The cohort was compiled using the Finnish population information system, an electronic register including personal data of all permanent residents in Finland, such as name, gender, the unique and permanent Finnish personal identity code, date of birth and death. The study cohort consists of individuals alive on January 1, 2020.

### Exposed and unexposed follow-up periods

The primary outcome of interest was *confirmed CVST with thrombocytopenia*. Secondary outcomes were *register-based CVST* and *confirmed CVST*. The primary exposures of interest were the first doses of each of the COVID-19 vaccinations. A secondary exposure of interest was a confirmed COVID-19 disease.

The first 28 days following a BNT162b2 (Pfizer–BioNTech), mRNA-1273 (Moderna), or a ChAdOx1 nCov-19 vaccine dose served as risk periods and time before any of these vaccinations provided the unexposed follow-up time. Exposures to COVID-19 vaccine brands other than ones displayed were extremely rare. All comparisons related to vaccinations exclude subjects under age 16, since COVID-19 vaccines were unavailable for them at the time of the study. We also excluded the time after the vaccination risk periods from the analyses. To

enable clinically relevant comparisons between the risk and control periods, we express incidence per 28 days per million persons (for incidences per 100,000 person years, py, see the Supplement). In our separate independent analysis of exposure to COVID-19 infection we applied a similar approach.

For each individual in the cohort, the follow-up for CVST outcomes lasted from January 1st, 2020, until April 2, 2021, death, or the end of the first risk period for the evaluated exposure (COVID-19 vaccine or infection), whichever came first. The exposure dates were the date of vaccination or the sample date of a positive PCR test for the COVID-19 infection. We were unable to systematically collect data on symptom onset. The date of the first contact to hospital resulting in a relevant CVST diagnosis, with or without admission, indicated disease episode onset, and a new CVST episode could start at the earliest 28 days after visit / discharge. We included all relevant data provided in care notifications sent from the caregivers to the register, from the first visit to the hospital (usually emergency room) and/or admission (episode start) until discharge.

### Register-based CVST

We obtained information on hospital emergency-room visits and non-scheduled in-patient hospitalizations assigned ICD-10 code I636/I676/G08 either as the primary or any secondary diagnosis from the National Care Register for Health Care [13], which covers all in- and out-patient care provided in the Finnish hospitals.

### Confirmed CVST

To confirm the register-based CVST diagnoses, we retrieved all patient records of the register-based CVST cases from 6 months prior to the first hospital contact until April 2, 2021 from the mandatory nationwide electronic Kanta Patient Data Repository, in which all patient data recorded by the health care units are archived [14]. Three clinically experienced physicians, two pediatricians and one epidemiologist with a generalist background (P.H., A.A.P. and T.K.) independently reviewed all cases (two reviewers per case) and classified them either as a *confirmed CVST or no CVST*, on the basis of clinical radiological reports and clinical interpretations by the treating physician. Neuroimaging results including angiography by either computerized tomography, magnetic resonance imaging, or both, were available for all scrutinized episodes. All three reviewers together sought consensus in case of discrepancy between the two independent reviews. Original radiological images were not reviewed.

### Confirmed CVST with thrombocytopenia

We extracted the platelet count results from the Kanta Patient Data Repository. A platelet count below 150,000 per cubic millimeter from 14 days before through 14 days after the confirmed CVST episode start indicated thrombocytopenia. One expert hematologist (R.L.) confirmed the episodes without a chronic disease as a potential etiology for thrombocytopenia.

### SARS CoV-2 infections

All laboratory-confirmed SARS-CoV-2 infections are recorded in the National Infectious Diseases Register [15]. During the study period, test positivity with absence of positive tests during the proceeding 12 months indicated a distinct COVID-19 case. Real-time reverse-transcription polymerase chain reaction (PCR) (involved in 97.7% of all cases) was the most common test, while 2.3% of the cases based solely on the antigen detection method. The testing capacity exceeded 3,000 daily samples per million inhabitants since August, 2020 [16].

## COVID-19 vaccinations

Municipalities are responsible for organizing COVID-19 vaccinations in Finland. Vaccinations started December 27, 2020, with the BNT162b2 vaccine. Priority was first given to the elderly (70 and above), health care personnel (HCP) treating COVID-19 patients, and patients and HCP in long-term facilities. Persons under 70 in order of decreasing age groups followed next. On January 20, 2021, mRNA-1273 vaccine was introduced, and on Feb 10, 2021, ChAdOx1 nCov-19 vaccine with the priority to high-risk subjects less than 65 years of age. On March 19, 2021, the ChAdOx1 nCov-19 vaccine was put on hold and its use was resumed on March 29 but recommended only to the elderly ($\geq$65 years of age).

All vaccinations in Finland are recorded in the local electronic health care databases, from which they are automatically transferred overnight to the Primary Health Care Visits Register and extracted to the National Vaccination Register [17] (S2 Table). The data include unique personal identity codes, brand names and dates of the vaccinations. The analysis focused on the first doses only, due to the low number of 2nd doses administered by the time ChAdOX1 nCov-19 was put on hold.

## Chronic concomitant disease

To characterize the CVST cases, we extracted relevant diagnostic data on comorbidities from National Care Register for Health Care and from the Reimbursements for medical expenses register [18]. All issued prescriptions are archived electronically in the Prescription Centre [19]. We utilized these from Jan 1, 2014, to Nov 30, 2020. (For details, see Salo H et al. [20] and S3 Table).

Personal data obtained from different sources were linked via the unique personal identity codes.

## Statistical analyses

We express the results as ratios of two Poisson rates, where the incidence during the risk periods following exposure are compared with the incidence in the unexposed state. We considered individuals as unexposed until the day before the first observed exposure. Number of episodes regarding each CVST outcome divided by the person time the cohort spent in the respective states (exposed/unexposed) provided the absolute incidences. The 28-day adjusted incidence in the exposed minus the adjusted incidence without exposure during the same follow-up time provided the exposure-attributable risk, the reciprocal of which is the number needed to harm.

For adjusted analysis, we utilized Poisson regression and treated the exposures as time-dependent covariates indicating transition from the unexposed state to an exposed state during the follow-up. We aggregated the individual follow-up data by counting the person time and outcome events by exposure state, sex and age (at the turn of 2020/2021) and used the following model:

$$\log(\lambda_i) = \alpha + \beta_1 Exposure_i + \beta_2 Agegroup_i + \beta_3 Sex_i,$$

where $\lambda_i = E[Y_i]/t_i$ is the incidence in group $i$, given the expected number of outcome events $E[Y_i]$ and person time $t_i$ (where $Y_i \sim$ Poisson $(\lambda_i t_i)$), *Exposure* is an indicator for the exposure state, *Agegroup* abuses notation and denotes indicators for different age groups (grouped as in Table 1), *Sex* is an indicator for gender, and $\alpha$ is the intercept. We used Bayesian methods for parameter estimation, see the appendix for details. We report credible intervals for the exponent of $\beta_1$, which estimates the incidence rate ratio between the exposed and unexposed states.

**Table 1. Incidences per 28 days per million persons of cerebral venous sinus thrombosis and thrombocytopenia during the 28-day risk periods after COVID-19 vaccinations and during the unexposed time preceding the vaccinations.** Numbers of episodes provided in parentheses.

| Case Definition and age group, years | Unexposed time | | | BNT162b2 | | | ChAdOx1 nCov-19 | | |
|---|---|---|---|---|---|---|---|---|---|
| | Males | Females | Both sexes | Males | Females | Both sexes | Males | Females | Both sexes |
| CVST, register-based[a] | | | | | | | | | |
| 0–15 | 0.27 (2) | 0.71 (5) | 0.49 (7) | NA | NA | NA | NA | NA | NA |
| 16–29 | 0.79 (6) | 1.68 (12) | 1.22 (18) | 0 | 0 | 0 | 0 | 230.21 (1) | 153.63 (1) |
| 30–54 | 1.30 (19) | 2.18 (30) | 1.73 (49) | 0 | 18.37 (1) | 14.59 (1) | 58.92 (1) | 0 | 24.99 (1) |
| 55–64 | 2.73 (16) | 2.71 (16) | 2.72 (32) | 0 | 0 | 0 | 0 | 0 | 0 |
| 65+ | 2.12 (19) | 3.12 (35) | 2.68 (54) | 0 | 3.95 (1) | 2.41 (1) | 0 | 0 | 0 |
| All ages | 1.40 (62) | 2.18 (98) | 1.79 (160) | 0 | 5.72 (2) | 3.73 (2) | 12.30 (1) | 11.99 (1) | 12.14 (2) |
| CVST, confirmed[b] | | | | | | | | | |
| 0–15 | 0.14 (1) | 0.43 (3) | 0.28 (4) | NA | NA | NA | NA | NA | NA |
| 16–29 | 0.66 (5) | 1.40 (10) | 1.02 (15) | 0 | 0 | 0 | 0 | 230.21 (1) | 153.63 (1) |
| 30–54 | 1.09 (16) | 1.38 (19) | 1.23 (35) | 0 | 0 | 0 | 58.92 (1) | 0 | 24.99 (1) |
| 55–64 | 2.05 (12) | 1.69 (10) | 1.87 (22) | 0 | 0 | 0 | 0 | 0 | 0 |
| 65+ | 1.57 (14) | 2.14 (24) | 1.88 (38) | 0 | 0 | 0 | 0 | 0 | 0 |
| All ages | 1.08 (48) | 1.47 (66) | 1.28 (114) | 0 | 0 | 0 | 12.30 (1) | 11.99 (1) | 12.14 (2) |
| CVST, confirmed, with thrombocytopenia[c] | | | | | | | | | |
| 0–15 | 0.14 (1) | 0.28 (2) | 0.21 (3) | NA | NA | NA | NA | NA | NA |
| 16–29 | 0.13 (1) | 0 | 0.07 (1) | 0 | 0 | 0 | 0 | 230.21 (1) | 153.63 (1) |
| 30–54 | 0.07 (1) | 0 | 0.04 (1) | 0 | 0 | 0 | 58.92 (1) | 0 | 24.99 (1) |
| 55–64 | 0.68 (4) | 0.34 (2) | 0.51 (6) | 0 | 0 | 0 | 0 | 0 | 0 |
| 65+ | 0.45 (4) | 0.09 (1) | 0.25 (5) | 0 | 0 | 0 | 0 | 0 | 0 |
| All ages | 0.25 (11) | 0.11 (5) | 0.18 (16) | 0 | 0 | 0 | 12.30 (1) | 11.99 (1) | 12.14 (2) |

ChAdOx1 nCov-19 (Vaxzevria, AstraZeneca) BNT162b2 (Comirnaty, Pfizer–BioNTech). NA not applicable, for those under 16 years the COVID-19 vaccines were unavailable.

[a] CVST, cerebral venous sinus thrombosis: As a main diagnosis any of ICD-10 codes I636, I676, or G08 included. Only emergency-room visits and non-scheduled in-patient hospitalizations were included.

[b] CVST, confirmed: Episodes in registers that were confirmed by chart review (clinical radiological reports and clinical interpretations).

[c] CVST, confirmed, with thrombocytopenia: Platelet count < 150,000 per cubic millimetre within 14 days before and after episode start.

Note that the 28-day risk time after BNT162b2 was free from episodes with confirmed CVST. Note also, that risk time after mRNA-1273 (Moderna) was free from any CVST.

For incidences per 100,000 per year, see S6 Table.

**Ethics.** According to the Finnish Communicable Diseases Act, the Finnish Institute for Health and Welfare (THL) is in charge of monitoring the impact including effectiveness and safety of vaccines used in the national vaccination program and taking measures to investigate potential adverse events suspected to be linked with vaccination [21]. THL has the statutory right, notwithstanding confidentiality provisions, to access necessary information in patient documents and to link this information with other relevant register data. Written consent was obtained from patients or their next-of-kin, for the two individuals whose case reports are presented (S4 Table).

## Results

Altogether, we detected 167 *register-based CVST* episodes in 156 subjects (Fig 1). Of these, 50 episodes in 41 patients were not novel cases of a radiologically confirmed CVST in our patient file review. The remaining 117 (70%) in 115 patients were classified as *confirmed CVST*.

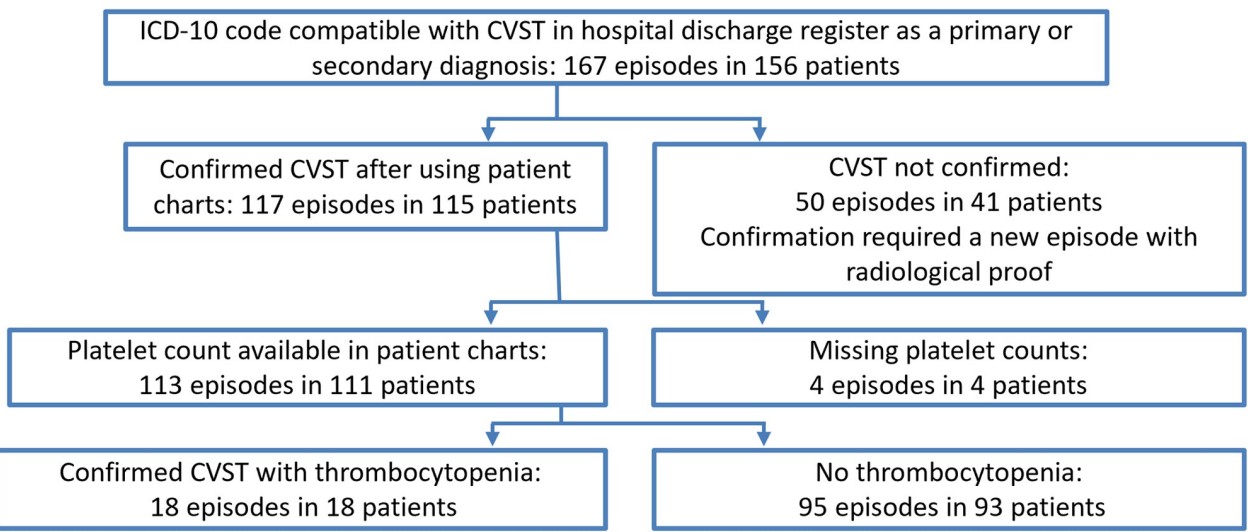

**Fig 1. Flow chart.** From top to bottom shows the extraction of confirmed cerebral venous sinus thrombosis (CVST) in combination with thrombocytopenia. Note that three register-based CVST episodes, one of which confirmed, are included here but occurred after the risk periods and were not included in the pre-vaccinated time.

Platelet counts were available for 113 of the 117 (97%) *confirmed CVST* episodes and in 18 of them, we detected a minimum platelet count less than 150,000 per cubic millimeter. None of the 18 had pre-existing thrombocytopenia or comorbidity explaining the thrombocytopenia. During 3 of the 18 episodes, acute concomitant disease likely to contribute to the thrombocytopenia was diagnosed (infection, operation and head trauma, S7 Table). No *confirmed CVST* episodes emerged during 28 days after or at any time after positive testing indicating COVID-19 infection (for infection data, see S5 Table).

### Episodes prior to COVID-19 vaccination

For the unexposed follow-up time prior to vaccination, the incidences of *register-based CVST* and *confirmed CVST* were 1.8 and 1.3 per 28 days per million persons (n = 160 and 114, Table 1, for values per 100,000 py, see S6 Table). Incidence of *confirmed CVST with thrombocytopenia* was 0.18 per 28 days per million persons. The incidence of *confirmed CVST* was quite stable in various age groups above 30 with a minor female predominance, whereas the incidence of *confirmed CVST with thrombocytopenia* was very low in age groups 16 to 54 years, peaked at 55 to 64 years and had a male predominance.

Three *confirmed CVST* episodes (2.6%) resulted in death within 30 days from the start of the episode (Table 2). None of these patients had thrombocytopenia. The median length of hospitalization was 14 and 7 days in those with and without thrombocytopenia, respectively. In the registers, 11 (9.7%) of the *confirmed CVST* episodes, were associated with ICD-10 codes indicating intracranial surgery, trauma or tumor. Malignancies were more common among those without (16.1%) than with (6.3%) thrombocytopenia.

### Episodes following COVID-19 vaccination

By the end of the follow-up, 200,397 first vaccinations were given with ChAdOx1 nCov-19, 714,727 with BNT162b2 and 67,877 with mRNA-1273 (S2 and S5 Tables). Of the four *register-based CVST* episodes emerging within 28 days after COVID-19 vaccinations, the two that

**Table 2. Characteristics and course of illness concerning patients and episodes of cerebral venous sinus thrombosis in unvaccinated subjects.**

| Group | CVST from register data | Confirmed CVST | Confirmed CVST with thrombocytopenia |
|---|---|---|---|
| | n = 150 patients, 160 episodes | n = 112 patients, 114 episodes | n = 16 patients, 16 episodes |
| **Patient characteristics, by patient** | | | |
| Female sex, count, percent | 92 (61.3%) | 65 (58.0%) | 5 (31.3%) |
| Age, median (years) | 58 | 58 | 61 |
| Chronic concomitant diseases[a] count, percent | | | |
| Type 2 Diabetes | 12 (8.0%) | 7 (6.3%) | 2 (12.5%) |
| Malignancy | 24 (16.0%) | 18 (16.1%) | 1 (6.3%) |
| Cardiovascular disease | 29 (19.3%) | 20 (17.9%) | 6 (37.5%) |
| Severe lung disease | 8 (5.3%) | 6 (5.4%) | 1 (6.3%) |
| Other risk group[b] | 61 (40.7%) | 44 (39.3%) | 6 (37.5%) |
| None of the above | 89 (59.3%) | 68 (60.7%) | 10 (62.5%) |
| **Course of acute illness, by episode** | | | |
| Length of stay in acute care hospital, days, median (range) | 5 (0–141) | 7 (0–141) | 13.5 (3–141) |
| Death within 30 days of episode start/hospitalization, n (%) | 3 (1.9%) | 3 (2.6%) | 0 (0%) |
| Cerebral infarction, n (%)[c] | 47 (29.4%) | 29 (25.4%) | 5 (31.3%) |
| Cranial surgery, trauma or tumor, count, percent[c] | 11 (6.9%) | 11 (9.7%) | 1 (6.3%) |
| Platelet count nadir ± 14 days of episode start, per cubic mm, median (range)[f] | 225,000 (14,000–647,000) | 223,000 (14,000–647,000) | 110,000 (14,000–146,000) |

Abbreviations: CVST, cerebral venous sinus thrombosis. For ICD-10 and other codes, see S3 Table.

[a] Chronic concomitant diseases from the Care register for Health Care, Register for drug reimbursements and Register for Drug purchases. From Jan 1, 2015 to Jan 1, 2021 or to fist CVST episode.

[b] Other risk groups for vaccination: Diabetes with nephropathy, Severe chronic kidney disease, History of transplantation, Down syndrome, Congenital immunodeficiency, Sequelae of head trauma, Chronic severe liver disease, Other diabetes, Adrenal disorders, Sleep apnea, Clozapine therapy, Schizophrenia, Muscle disorders (S3 Table).

[c] Cerebral infarction, cranial surgery, trauma or tumor, during episode.

[f] For the 98 episodes with Confirmed CVST but without thrombocytopenia these where 239,000 (156,000–647,000) and the 1st and 3rd quartile limits where 205,000 and 286,000.

were confirmed to be CVST also coincided with thrombocytopenia, in a man aged 39 and a woman aged 21, both after ChAdOx1 nCov-19 vaccination (Table 1 and S4 Table). The male deceased 5 days after vaccination.

No confirmed CVST episodes occurred within 28 days after about 780,000 mRNA vaccinations. No *CVST with thrombocytopenia* episodes emerged at any time after mRNA vaccinations. Three *register-based CVST episodes without thrombocytopenia* occurred after the predefined risk period and were consequently excluded from the analyses. One of them was a *confirmed CVST* episode and occurred right after the risk period following a BNT162b2 vaccination.

## Comparing incidences with and without exposure to COVID-19 vaccinations

Within 28 days after the ChAdOx1 nCov-19 vaccine the crude incidence of CVST with thrombocytopenia was 12.1 per 28 days per million persons, 67-fold the background rate. Our adjusted model accounting for age and sex resulted in a rate ratio of 40 (median, 95% credible

interval 6 to 161). The corresponding adjusted number needed to harm was 104,000 (median, 95% credible interval 31,000 to 848,000).

## Discussion

We found the population-based incidence of CVST coinciding with thrombocytopenia to be considerably lower than for any CVST, 0.23 (95% CI, 0.13 to 0.38) vs. 1.66 (1.37 to 2.00) per 100,000 person-years. Our findings strengthen the interpretation that the incidence of CVST with thrombocytopenia after ChAdOx1 nCov-19 vaccination clearly exceeds the expected.

Background incidences of possible adverse effects following COVID-19 vaccinations are important for monitoring the impact of vaccination program [22]. The term vaccine-induced immune thrombotic thrombocytopenia (VITT) has gained widespread use for TTS following vaccination since its recognition in spring 2021 [23].

We are aware of one report of population-based incidence rates of CVST with thrombocytopenia [24]. Background rates for CVST with thrombocytopenia (0.1, 95% CI, 0.1 to 0.2, per 100,000 person-years) were similar to ours. A hospital-based multicentre study indicated that baseline thrombocytopenia occurs in 8% of CVST patients [12]. In our population-based study, 14% of those with *confirmed CVST* had thrombocytopenia at any time from 14 days before to 14 days after admission.

Our setting brings additional information to the background rates that already are published. A large self- controlled case series study in the UK showed moderate vaccine-related rise in incidence rates for venous thromboembolism, thrombocytopenia and their combination, as based on diagnoses [25]. Access to laboratory data enabled us to detect thrombocytopenia more comprehensively. We performed a chart review on all cases. In our setting, only 71% (114/160) of the register-based CVST cases during unexposed time remained CVST after chart review.

Incidence of confirmed CVST was 1.7/100,000 py. This finding aligns with previous reports in adult populations:1.3/100,000 in the UK [25], 1.3/100,000 py in the Netherlands [26], and 1.6/100,000 py in Adeleide, Australia [27], and 1.3/100,000 py in Finland [28]. In our data, 2 of the 117 patients presented with more than a single episode during the study period. CVST is associated with mortality of 0 to 2%, according to a recent review showing a declining trend [29]. CVST represents the most common clinical presentation of TTS with considerably higher mortality exceeding 26 to 56% in reported post-vaccination cases (S1 Table). In concordance, our estimate for 30-day mortality in *confirmed CVST* patients was 2.6%, whereas one out of the two vaccinated patients with *confirmed CVST with thrombocytopenia* in our data deceased. However, all 16 unvaccinated patients with *confirmed CVST and thrombocytopenia* survived 30 days. The hospital-based report with 73 such cases or the Catalonian manuscript did not include mortality data [12, 24]. For vaccinated subjects, however, TTS with CVST mortality is estimated to be 38% in a recent review [8].

The full spectrum of TTS includes thrombosis in sites such as splanchnic and portal veins also. Instead for focusing on those forms of TTS, we focused on cerebral vein sinus thrombosis in our analysis, because of its high mortality, straightforward ICD-10 coding, and clear-cut radiology reports.

COVID-19 itself is associated with both thrombosis and thrombocytopenia [30, 31]. In a UK study of vaccinated subjects, 1.8 million COVID-19 infections were associated with higher rates of thromboembolic outcomes, including CVST, the excess rate of which was 2 per million from day 8 to 28 [25]. None of our 73,000 COVID-19 infected patients developed radiologically confirmed CVST within 28 days of follow-up.

A Nordic collaboration study revealed a clear excess of venous thromboembolic events following the ChAdOx1 nCov-19 vaccinations [32]. The excess rate for CVST was 7 per million

vaccinated, which is in line with our crude excess rate of 12 per million persons per 28 days. Notably, confirming the findings of others [33, 34], the 28-day risk periods following more than 700,000 mRNA COVID-19 vaccinations were likewise free from radiologically confirmed CVST.

We present the incidence of *confirmed CVST with thrombocytopenia* in the whole population of Finland with comprehensive methods that minimize detection bias. Reporting to the Care Register for Health Care is compulsory and covers the whole country. Also, socioeconomic factors are unlikely to influence the detection since there is universal access to tertiary health care, almost free of charge.

Our study has weaknesses. As any other vaccination surveillance study, also ours may be biased by the increase of public attention on an adverse effect. CVST typically presents with severe symptoms including intense and protracted headache. However, atypical CVST cases with mild symptoms may have been missed. Nevertheless, the CVST cases reported after adenoviral vector vaccination have had severe and clear symptoms. It is unlikely that these kinds of severe cases would be missed in our setting, irrespective of the vaccination status. Also, our definitions were based on clinicians' decisions and routine evaluation including the radiology. This might decrease precision, but only slightly, because of Finland's setting with equal access to our high-quality health care system.

Our vaccination-status data was recorded at the time of administration, independently of the outcome data. Individual patient data for all cases found in the Care Register for Health Care were retrieved and more than 98% of the confirmed cases were linkable to timely platelet count data.

We found 2 cases of CVST with thrombocytopenia after vaccination of 200,000 subjects with ChAdOx1 nCov-19 adeno-virus-vector vaccine. In our analysis adjusted for age and sex, the incidence rate ratio versus those unvaccinated was 40 (median, 95% credible interval 6 to 161). Since the vaccine was withdrawn from subjects younger than 65, a longer catchment period would hardly have increased the number of vaccinated cases or the limited precision of our vaccine adverse effect analysis.

Due to the vaccination prioritisation, most ChAdOx1 nCov-19 adeno-virus-vector vaccine doses were administered to subjects with comorbidities, which raises the question of bias in the risk estimates. If such comorbidities predisposed to CVST with thrombocytopenia, this would result in overestimation of vaccine-attributable risk. Yet, a considerably higher estimate has been reported from Norway in health care professionals [2].

Several COVID-19 vaccines have been, and are being, developed with different technologies, including two adenoviral vector vaccines, ChAdOx1 nCov-19 and Ad26.COV2.S. Also the latter has been associated with TTS, yet with lower incidence. Availability of various vaccines by several manufacturers is needed to fight the epidemic globally. One needs to balance between the pros and cons of each formulation. Accurate data, both on effectiveness and safety, are needed to evaluate the best options for each setting. In this rapidly developing field, the retrospective observations will guide us on safety. Meanwhile, for reliable safety-monitoring, research on background incidences in valid settings, such as ours, is crucial.

## Supporting information

**S1 Table. Thrombosis with thrombocytopenia cases reported following ChAdOx1 nCov-19 vaccination (AstraZeneca).**
(DOCX)

**S2 Table. Administered doses by Apr 2, 2021 (end of follow-up time) of each vaccine and number of COVID-19 infections by age and risk group.**
(DOCX)

**S3 Table. Codes for diseases in registers.**
(DOCX)

**S4 Table. Case report.**
(DOCX)

**S5 Table. Total population by exposure, age group and sex.**
(DOCX)

**S6 Table. Incidences per 100,000 person years of cerebral venous sinus thrombosis and thrombocytopenia during unexposed time.** Estimates and 95% confidence intervals.
(DOCX)

**S7 Table. Possible etiology of the confirmed cerebral venous sinus thrombosis with thrombocytopenia cases.**
(DOCX)

**S1 Text. Supplemental methods.** A detailed description of Bayesian estimation methodology.
(DOCX)

## Acknowledgments

We thank Heini Salo and Tuija Leino, from the Finnish Institute for Health and Welfare (THL), for their contributions of preparing the list of disease-based risk groups regarding the COVID-19 vaccination program in Finland. We also thank the data management team, especially Timo Koskenniemi, THL for development of data-processing tools required to utilize the data in health records.

## Author Contributions

**Conceptualization:** Petteri Hovi, Arto A. Palmu, Tuomo A. Nieminen, Miia Artama, Jukka Jokinen, Esa Ruokokoski, Hanna Nohynek, Terhi Kilpi.

**Data curation:** Tuomo A. Nieminen, Jukka Jokinen, Esa Ruokokoski.

**Formal analysis:** Tuomo A. Nieminen, Esa Ruokokoski.

**Funding acquisition:** Miia Artama.

**Investigation:** Petteri Hovi, Arto A. Palmu, Tuomo A. Nieminen, Miia Artama, Riitta Lassila, Hanna Nohynek, Terhi Kilpi.

**Methodology:** Petteri Hovi, Arto A. Palmu, Tuomo A. Nieminen, Miia Artama, Jukka Jokinen, Esa Ruokokoski, Riitta Lassila, Hanna Nohynek, Terhi Kilpi.

**Project administration:** Petteri Hovi, Arto A. Palmu, Hanna Nohynek, Terhi Kilpi.

**Supervision:** Petteri Hovi, Arto A. Palmu, Jukka Jokinen, Hanna Nohynek, Terhi Kilpi.

**Validation:** Riitta Lassila, Terhi Kilpi.

**Writing – original draft:** Petteri Hovi, Tuomo A. Nieminen, Miia Artama.

**Writing – review & editing:** Petteri Hovi, Arto A. Palmu, Tuomo A. Nieminen, Miia Artama, Jukka Jokinen, Esa Ruokokoski, Riitta Lassila, Hanna Nohynek, Terhi Kilpi.

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
