## [Decision Letter · Decision Letter 0]

8 Aug 2022

PONE-D-22-09809Incidence of Sinus Thrombosis with Thrombocytopenia - a Nation-wide Register StudyPLOS ONE

Dear Dr. Hovi,

Thank you for submitting your manuscript to PLOS ONE. After careful consideration, we feel that it has merit but does not fully meet PLOS ONE’s publication criteria as it currently stands. Therefore, we invite you to submit a revised version of the manuscript that addresses the points raised during the review process.

We look forward to receiving your revised manuscript.

Kind regards,

Roza Chaireti

Academic Editor

PLOS ONE

Journal Requirements:

The current study received no external funding outside THL and there are no conflicting interests. Finnish Institute for Health and Welfare (THL) conducts Public-Private Partnership with vaccine manufacturers and has received research funding from Sanofi Inc., Pfizer Inc., and GlaxoSmithKline Biologicals SA.  AAP, TN, JJ, ER, and TK have been investigators in these studies, but they have received no personal remuneration. P.H. reports being, in 2016 and 2017, a member in an expert group “Future Rheumatology Advisory Board”, funded by Pfizer.

The authors received no specific funding for this work

The current study received no external funding outside THL and there are no conflicting interests. Finnish Institute for Health and Welfare (THL) conducts Public-Private Partnership with vaccine manufacturers and has received research funding from Sanofi Inc., Pfizer Inc., and GlaxoSmithKline Biologicals SA.  AAP, TN, JJ, ER, and TK have been investigators in these studies, but they have received no personal remuneration. P.H. reports being, in 2016 and 2017, a member in an expert group “Future Rheumatology Advisory Board”, funded by Pfizer.

Reviewers' comments:

Reviewer's Responses to Questions

**Comments to the Author**

1. Is the manuscript technically sound, and do the data support the conclusions?

Reviewer #1: Yes

Reviewer #2: Yes

2. Has the statistical analysis been performed appropriately and rigorously? 

Reviewer #1: Yes

Reviewer #2: Yes

3. Have the authors made all data underlying the findings in their manuscript fully available?

Reviewer #1: Yes

Reviewer #2: Yes

4. Is the manuscript presented in an intelligible fashion and written in standard English?

Reviewer #1: Yes

Reviewer #2: Yes

5. Review Comments to the Author

Reviewer #1: The introduction of thrombosis in unusual regions, including cerebral venous sinus thrombosis(CVST), by SARS-CoV-2 vaccines, caused a major public concern. Although vaccine-induced CVST is rare, for instance, 0.55 per 100,000 person-months in the first month after 7,126,434 vaccination in Germany, the affliction of young and previously healthy people makes it a critical area of research.

 This is a cohort study to investigate the association between COVID-vaccination and the evolution of CVST. Authors showed that only 2 out of 117 patients with CVST fulfill the criteria of CVST associated with vaccine-induced thrombosis and thrombocytopenia(VITT). In the other word, the rate of CVST-VITT was about 1 per 100.00 adult population. 

Research design is sound. Methods are adequately described, and results are clearly presented. Some significant issues should be addressed, though:

Method:

-“ The date of the first contact to hospital resulting in a relevant CVST diagnosis indicated disease onset” seems to be obscure. The authors should clearly differentiate between the date of admission and the date of first clinical presentation ( the date of onset)

-“Original radiological images were not reviewed.”. it should be considered a drawback.

- It should be mentioned that using the nationwide registry supervised by reviewers decreased the risk of false positives; however, some false-negative cases, particularly mild cases of CVST with an isolated headache, might be overlooked.

-Were there any patients diagnosed only by antigen detection method rather than PCR?

Results

-“We detected 167 register-based CVST episodes in 156 subjects”. Seven percent risk of recurrence in a 15-month period of observation is unusual. Authors should justify this result.

-Table-2

Concomitant diseases, which were mentioned in table-2 are not the most important predisposing factors for CVST. Provocative predisposing factors for CVST include pregnancy, puerperium, OCPs, infections, head injury, and dehydration. Non-provocative factors include inherent conditions such as acquired or genetic thrombophilia, malignancies, hematologic diseases, and inflammatory diseases. Some of them, such as SLE, antiphospholipid antibody syndrome, leukemia, and some medications, can induce both CVST and thrombocytopenia; and should be excluded for diagnosing VITT. However, they were not mentioned in the manuscript.

- The status of anti-platelet factor 4(PF4) antibodies was mentioned

Discussion:

The discussion part is too long.

Limitations of the study were not mentioned.

Reviewer #2: Thrombosis with thrombocytopenia syndrome (TTS) has been reported after vaccination with the adenoviral vector based SARS-CoV-2 vaccines ChAdOx1 nCov-19 (Oxford–AstraZeneca) and Ad26.COV2.S (Janssen/Johnson & Johnson). Using nationwide registers, the authors estimated the incidence of CVT and CVT with thrombocytopenia since 2020 to established whether vaccines against SARS-CoV-2 were associated with increased risk of CVT with thrombocytopenia.

Comments and suggestions are listed below:

1. Thrombocytopenia is a key feature of the vaccine-induced immune thrombotic thromcytopenia syndrome (VITT). However, the authors have not addressed this in their analysis of ADRs. This should be acknowledged as a major limitation.

2. The authors say in the results that CVT occurring within 28 days was considered. However, this is not detailed in the methods. Case definition and adjudication should be better described in the methods section.

3. the number of events is small (only 2 patients with CVT and thrombocytopenia after vaccination, only 200.000 subjects vaccinated with ChAdOx 1 nCov-19). For this reason, there is substantial imprecision in the estimates that are provided (very large confidence intervals). The discussion on the study limitations should be expanded.

4. Three physicians revised all suspected clinical cases. Please explain whether these clinicians were neurologists or physicians with a special expertise in cerebrovascular diseases.

5. If possible, the authors should consider describing what were the causes/comorbidities in patients with thrombocytopenia and CVT, particularly those that may justify concomitant thrombocytopenia. Information on the range of platelet values would be also useful (less severe thrombocytopenia in patients without VITT?)

6. A more recent study which was based on pharmacovigilance data from the EEA (doi: 10.1212/WNL.0000000000013148) provides more updated information than the EMA report that has been cited in Supplementary table 1 (reference 6). Consider updating that.

7. Previous studies, such as doi: 10.1212/WNL.0000000000013148, have also not found an increased risk of CVT in recipients of mRNA vaccines against SARS-CoV-2. Consider discussing that.

8. No data is provided for Ad26.COV2.S. The reasons for this should be explained.

6. PLOS authors have the option to publish the peer review history of their article (what does this mean?). If published, this will include your full peer review and any attached files.

Reviewer #1: **Yes: **Afshin Borhani Haghighi

Reviewer #2: No

---

## [Author Response · Author response to Decision Letter 0]

26 Oct 2022

Please see the Response to the Reviewers.

---

## [Decision Letter · Decision Letter 1]

5 Jan 2023

PONE-D-22-09809R1Incidence of sinus thrombosis with thrombocytopenia: A nationwide register studyPLOS ONE

Dear Dr. Hovi,

Thank you for submitting your manuscript to PLOS ONE. After careful consideration, we feel that it has merit but does not fully meet PLOS ONE’s publication criteria as it currently stands. Therefore, we invite you to submit a revised version of the manuscript that addresses the points raised during the review process by one of the reviewers.

We look forward to receiving your revised manuscript.

Kind regards,

Roza Chaireti

Academic Editor

PLOS ONE

Reviewers' comments:

Reviewer's Responses to Questions

**Comments to the Author**

1. If the authors have adequately addressed your comments raised in a previous round of review and you feel that this manuscript is now acceptable for publication, you may indicate that here to bypass the “Comments to the Author” section, enter your conflict of interest statement in the “Confidential to Editor” section, and submit your "Accept" recommendation.

Reviewer #1: All comments have been addressed

Reviewer #2: All comments have been addressed

2. Is the manuscript technically sound, and do the data support the conclusions?

Reviewer #1: Yes

Reviewer #2: Yes

3. Has the statistical analysis been performed appropriately and rigorously? 

Reviewer #1: Yes

Reviewer #2: Yes

4. Have the authors made all data underlying the findings in their manuscript fully available?

Reviewer #1: Yes

Reviewer #2: No

5. Is the manuscript presented in an intelligible fashion and written in standard English?

Reviewer #1: Yes

Reviewer #2: Yes

6. Review Comments to the Author

Reviewer #1: Dear Editor,

I still have some concerns about the manuscript:

Response #1 – 5

The authors response is ambiguous. They should directly declare in the manuscript why

they had 167 register-based CVST episodes in 156 subjects in a 15-month period of observation in the first revision.

The changed manuscript “In our data, 2 of the 117 patients presented with more than a single [confirmed CVST] episode during the study period.” did not reply my argument

Response #1 – 6:

I agree that “We only detected two cases and were unable to perform formal analysis that take concomitant disease into account”. However, the authors should clearly declare in the method part that conditions which can induce both CVST and thrombocytopenia was excluded for diagnosing VITT.

Reviewer #2: Thank you for the revised manuscript and for your responses to each of the comments made in the previous review of this submission.

7. PLOS authors have the option to publish the peer review history of their article (what does this mean?). If published, this will include your full peer review and any attached files.

Reviewer #1: No

Reviewer #2: No

---

## [Author Response · Author response to Decision Letter 1]

8 Jan 2023

Please see the copy sent as the Word version. Tracked changes are not marked here.

“Reviewer #1: Dear Editor, 

 I still have some concerns about the manuscript: 

Response #1 – 5 

 The authors response is ambiguous. They should directly declare in the manuscript why 

 they had 167 register-based CVST episodes in 156 subjects in a 15-month period of observation in the first revision. 

 The changed manuscript ‘In our data, 2 of the 117 patients presented with more than a single [confirmed CVST] episode during the study period.’ did not reply my argument” 

Response #1 – 5 R1 08JAN2023. 

In Comment #1-5 on our primary submission, reviewer #1 argued that having 11 recurrencies after CVST all within the same 15-month recruitment period would be unusual. We agree. 

The reason is obvious: these 11 episodes were mostly not true recurrencies. We have now included more info in the manuscript right where the numbers 167 episodes in the 156 subjects are first stated. That is, we added some details of the register-based episodes that were not confirmed in the chart review. The seemingly high recurrence rate was true only for the register-based definition and, in fact, was not existent for the 117 episodes fulfilling the requirements of ‘confirmed CVST’. 

Our new addition (Results, row 170): 

“Altogether, we detected 167 register-based CVST episodes in 156 subjects (Fig 1). Of these, 50 episodes in 41 patients were not novel cases of a radiologically confirmed CVST in our patient file review. The remaining 117 (70%) in 115 patients were classified as confirmed CVST in our patient file review.” 

“Response #1 – 6: 

 I agree that ‘We only detected two cases and were unable to perform formal analysis that take concomitant disease into account’. However, the authors should clearly declare in the method part that conditions which can induce both CVST and thrombocytopenia was excluded for diagnosing VITT.” 

Response #1 – 6 R1 08JAN2023. 

Our definition of confirmed CVST with thrombocytopenia included a novel CVST with radiological confirmation and thrombocytopenia. This was done without excluding any cases with conditions that could simultaneously induce these two. As such, it fulfilled the requirements of the current Brighton collaboration TTS definition from Nov 2021, ref 7, available from: https://brightoncollaboration.us/thrombosis-with-thrombocytopenia-syndrome-interim-case-definition/. 

Although not excluding episodes with any a priori list of acute conditions that could induce thrombocytopenia we report them (see below). Please note that within our first resubmission, as a response to reviewer comment #2 -5, we also added more data on the 18 episodes with ‘confirmed CVST with thrombocytopenia’ (S8 Table). 

We now also refer to ‘S8 Table Possible etiology of the confirmed cerebral venous sinus thrombosis with thrombocytopenia cases’ from the text (Results, row 175): 

“During 3 of the 18 episodes, acute concomitant disease likely to contribute to the thrombocytopenia was diagnosed (infection, operation and head trauma, S8 Table).” 

We understand that in three instances, our wording “Concomitant disease” needs the word “chronic” to distinguish it from acute concomitant conditions and disease: 

Subheading in Methods, row 120 

Table 2, 3rd row 

Table 2, footnote a, row 199

---

## [Decision Letter · Decision Letter 2]

10 Feb 2023

Incidence of sinus thrombosis with thrombocytopenia: A nationwide register study

PONE-D-22-09809R2

Dear Dr. Hovi,

We’re pleased to inform you that your manuscript has been judged scientifically suitable for publication and will be formally accepted for publication once it meets all outstanding technical requirements.

Kind regards,

Roza Chaireti

Academic Editor

PLOS ONE

Additional Editor Comments (optional):

Reviewers' comments:

Reviewer's Responses to Questions

**Comments to the Author**

1. If the authors have adequately addressed your comments raised in a previous round of review and you feel that this manuscript is now acceptable for publication, you may indicate that here to bypass the “Comments to the Author” section, enter your conflict of interest statement in the “Confidential to Editor” section, and submit your "Accept" recommendation.

Reviewer #1: (No Response)

2. Is the manuscript technically sound, and do the data support the conclusions?

Reviewer #1: (No Response)

3. Has the statistical analysis been performed appropriately and rigorously? 

Reviewer #1: (No Response)

4. Have the authors made all data underlying the findings in their manuscript fully available?

Reviewer #1: (No Response)

5. Is the manuscript presented in an intelligible fashion and written in standard English?

Reviewer #1: (No Response)

6. Review Comments to the Author

Reviewer #1: (No Response)

7. PLOS authors have the option to publish the peer review history of their article (what does this mean?). If published, this will include your full peer review and any attached files.

Reviewer #1: **Yes: **Afshin Borhani-Haghighi

---

## [Editor Report · Acceptance letter]

15 Feb 2023

PONE-D-22-09809R2 

Incidence of Sinus Thrombosis with Thrombocytopenia - a Nation-wide Register Study 

Dear Dr. Hovi:

I'm pleased to inform you that your manuscript has been deemed suitable for publication in PLOS ONE. Congratulations! Your manuscript is now with our production department. 

Kind regards, 

on behalf of

Dr. Roza Chaireti 

Academic Editor

PLOS ONE